# A Bidirectional Loss Approach to Imparting Order Sensitivity to Multi-Image Chest X-ray Encoders

**Doowoong Choi**[1,2]                    JUSTICEJUSTICE314@KOREA.AC.KR
**Hanbin Ko**[2,3,4]                    LUCASKO1994@SNU.AC.KR
**Jung Hoon Lee**[2,3,4]                    CROP2292@SNU.AC.KR
**Chang Min Park**[2,5]                    MORPHIUS@SNU.AC.KR

[1] *Department of Computer Science and Engineering, Korea University.*
[2] *Department of Radiology, Seoul National University Hospital* [3] *Department of Interdisciplinary Program in Bioengineering, Seoul National University.* [4] *Integrated Major in Innovative Medical Science, Seoul National University Graduate School.* [5] *Institute of Medical and Biological Engineering, Seoul National University Medical Research Center, Seoul National University*

**Editors:** Accepted for publication at MIDL 2025

## Abstract

Longitudinal chest X-ray (CXR) analysis is a critical step in assessing disease progression, yet existing deep learning methods often fail to account for the inherent temporal directionality of serial images, yielding inconsistent predictions when their order is reversed. In this work, we propose a bidirectional loss framework that enforces order sensitivity in multi-image CXR encoders. Leveraging large language models (LLMs), we obtain fine-grained interval change labels—resolved, improved, stable, worsened, and new—by comparing prior and current radiology reports across five common thoracic findings. We then exploit the symmetric nature of these labels by reversing image order and inverting labels (e.g., improved - worsened) during training. Experiments on the MIMIC-CXR and CheXpert datasets show that our method surpasses baselines for most findings, effectively embedding order awareness while retaining a simple, efficient design.

**Keywords:** Deep Learning, Temporal Image Analysis, CXR

## 1. Introduction

Previous studies on CXR vision encoders have attempted to capture temporal dynamics by learning relationships between serial images (Oh et al., 2019) (Santeramo et al., 2018). **CheXRelNet**(Karwande et al., 2022) utilized 2-class labels {*improved, worsened*} to train models on disease progression between paired images. In a more fine-grained approach, **BioViL-T**(Bannur et al., 2023b) proposed a *temporal image classification* task, aiming to distinguish among {*improved, stable, worsened*} states. These studies highlight the growing interest in modeling temporal changes directly from serial chest X-rays, yet many fail to consider the *temporal direction*, producing inconsistent outputs when the input order is reversed. For example, if a model predicts "improved" when going from image A to B, it should predict "worsened" when going from B to A.

We address this by introducing a *bidirectional training approach* that enforces *order sensitivity*. Our key contribution involves computing loss from both original and temporally reversed image pairs, applying inverted labels (e.g., *new ↔ resolved*). To obtain high-quality interval labels, we use LLMs to analyze prior and current radiology reports and classify finding-level changes across five thoracic conditions. Experiments on MIMIC-CXR (Johnson et al., 2016) and CheXpert (Irvin et al., 2019)(Chambon et al., 2024) show consistently improved performance over baseline models.

## 2. Method

Our model takes a pair of CXRs (prior and current) as input and outputs a multi-label classification across the five findings, predicting one of five interval change categories. The input image pair is encoded using a transformer-based backbone and then passed through a task-specific classification head. BioVil-T encoder was used as backbone for the experiment.

**Label Extraction with LLMs:** Given a prior and current report, the LLM identified whether each of five common findings—*pleural effusion*, *pneumothorax*, *consolidation*, *edema*, and *pneumonia*—is present in either report. If a finding is present in at least one report, the model determines its interval status as one of the following: *resolved*, *improved*, *stable*, *worsened*, or *new* (Appendix A).

**Bidirectional Label Inversion:** We define the symmetric label mapping as follows:

$$new \leftrightarrow resolved, \quad improved \leftrightarrow worsened, \quad stable \leftrightarrow stable$$

With this mapping, we generate an additional training instance for each image pair by reversing their temporal order and applying the corresponding label inversion. This effectively doubles the size of the dataset, helping to mitigate the common issue of data scarcity in temporal image classification.

**Bidirectional Loss:** Let $x_p$ and $x_c$ denote the prior and current CXRs, and let $y$ denote the interval change label vector for all findings. We define the loss for the forward direction (i.e., *Unidirectional Loss*) as $\mathcal{L}(x_p, x_c, y)$. For the reversed input $(x_c, x_p)$, we apply the inverted labels $y'$ and compute the loss $\mathcal{L}(x_c, x_p, y')$. The final bidirectional loss is:

$$\mathcal{L}_{\text{bi}} = \mathcal{L}(x_p, x_c, y) + \mathcal{L}(x_c, x_p, y') \tag{1}$$

## 3. Result

| Method | P. Effusion | Edema | Pneumonia | Pneumothorax | Consolidation |
|---|---|---|---|---|---|
| Baseline | 46.2 | 53.4 | 13.9 | 13.9 | 15.2 |
| Unidirectional | 60.3 | 56.0 | 6.3 | 6.3 | 49.3 |
| Bidirectional | **81.8** | **66.8** | **57.4** | **57.4** | **73.0** |

Table 1: Inversion Consistency Rate derived from MS-CXR-T

**Order Sensitivity Validation with Reversed Inputs:** We introduce the Inversion Consistency Rate (ICR) metric, defined as the ratio of prediction pairs $(y, y')$ where the reversed prediction $y'$ obtained by swapping the input order, matches the expected inverse of the original prediction $y$ (e.g., worsen $\leftrightarrow$ improved)(Appendix B). With MS-CXR-T(Bannur et al., 2023a), consistently higher ICR was achieved using bidirectional method, which indicates that our model has acquired *order sensitivity*, avoiding inconsistent outputs when the input order is reversed.

Table 2: Comparison of model performance (F1, AUC, Accuracy) across five conditions. * samples with inter-radiologist disagreement were removed. No disagreement cases were found for Edema, Pneumonia, or Pneumothorax.

| Dataset | Method | P. Effusion | | | Edema | | | Pneumonia | | | Pneumothorax | | | Consolidation | | |
|---|---|---|---|---|---|---|---|---|---|---|---|---|---|---|---|---|
| | | F1 | AUC | Acc | F1 | AUC | Acc | F1 | AUC | Acc | F1 | AUC | Acc | F1 | AUC | Acc |
| MIMIC (5-class) | Uni | 42.7 | **78.9** | 47.4 | 38.5 | 73.3 | 40.1 | 21.7 | **77.7** | 49.2 | 17.7 | **54.7** | 33.9 | 39.8 | **75.3** | 47.4 |
| | Bi | **45.7** | 78.8 | **49.5** | **40.4** | **74.2** | **40.2** | **33.2** | 72.1 | **53.5** | **20.8** | 53.5 | **37.1** | **42.1** | 74.4 | **50.3** |
| MS-CXR-T (3-class) | Baseline | 44.0 | 70.4 | 46.7 | 49.4 | 75.5 | 57.9 | 29.9 | 60.5 | 63.2 | 24.2 | 39.9 | 23.7 | **45.6** | 67.2 | 51.7 |
| | Uni | 43.3 | 69.8 | 50.1 | 56.2 | 75.3 | 57.5 | 26.5 | 62.1 | **65.8** | 29.0 | **52.6** | 40.3 | 43.0 | 67.5 | **52.2** |
| | Bi | **50.0** | **72.6** | **54.0** | **58.2** | **77.2** | **59.3** | **43.8** | **69.4** | 59.5 | **29.3** | 49.4 | **44.1** | 40.5 | **70.7** | 50.2 |
| MS-CXR-T-X2 (3-class) | Baseline | 44.5 | 69.8 | 44.8 | 50.0 | 74.0 | **57.3** | 28.3 | 64.4 | 42.4 | 23.0 | 50.9 | 25.8 | 44.9 | 67.2 | 51.7 |
| | Uni | 45.4 | 69.3 | 49.9 | 53.6 | 73.2 | 54.5 | 23.3 | 66.1 | 40.0 | 27.6 | 46.9 | 40.5 | 41.4 | 67.5 | **52.2** |
| | Bi | **52.4** | **72.3** | **55.7** | **56.5** | **75.9** | 56.8 | **49.8** | **72.5** | **54.9** | **33.4** | **51.0** | **45.9** | **45.3** | **70.7** | 50.2 |
| MS-CXR-T + Curation* | Baseline | 44.2 | 72.2 | 46.6 | - | - | - | - | - | - | - | - | - | **46.3** | 67.5 | **53.2** |
| | Uni | 47.1 | 72.2 | 53.0 | - | - | - | - | - | - | - | - | - | 44.1 | 67.4 | 52.7 |
| | Bi | **57.3** | **75.6** | **60.2** | - | - | - | - | - | - | - | - | - | 41.4 | **70.6** | 50.3 |
| MS-CXR-T-X2 + Curation* | Baseline | 47.1 | 72.5 | 47.1 | - | - | - | - | - | - | - | - | - | **45.8** | 68.6 | 46.2 |
| | Uni | 50.3 | 72.5 | 53.6 | - | - | - | - | - | - | - | - | - | 41.8 | 68.0 | 49.1 |
| | Bi | **59.0** | **75.8** | **61.3** | - | - | - | - | - | - | - | - | - | 44.5 | **70.6** | **51.2** |
| CheXpert (external) | Baseline | 45.6 | **68.4** | 47.8 | 29.2 | 65.8 | 31.3 | 28.5 | 61.0 | **39.3** | 15.7 | 54.6 | 20.2 | 39.5 | 65.7 | 41.0 |
| | Uni | 48.6 | 65.4 | 50.2 | 38.9 | 65.4 | 38.9 | 27.0 | 61.7 | 39.1 | 27.3 | 54.5 | **40.7** | 44.8 | 64.0 | 50.0 |
| | Bi | **49.0** | 68.3 | **51.2** | **46.0** | **68.2** | **46.5** | **34.3** | **64.8** | 38.9 | **39.1** | **60.4** | 39.6 | **46.6** | **66.6** | **52.0** |

We evaluated our method(Bi) on four datasets, measuring F1-score, AUROC, and accuracy for each finding individually. F1 and AUC were measured with a one-vs-all setup and then macro-averaged, while accuracy represents the overall classification accuracy. As the Unidirectional method (Uni) replaces the baseline 3-class classifier with a 5-class head, it enables evaluation of whether increased label granularity improves performance.

**Consistent F1 Gains:** Bidirectional method consistently outperforms unidirectional method in MIMIC(5-class) dataset (Appendix C)and MS-CXR-T(w/o consolidation) in terms of F1-score. These results suggest that the bidirectional loss encourages better precision-recall balance in real-world clinical settings despite its simplicity. Inconsistent results in consolidation dataset warrant further analysis (Appendix D).

**Order Sensitivity:** MS-CXR-T-X2 augments the original MS-CXR-T dataset with reversed image sequences and corresponding inverted labels. Consistent improvements in both F1 and AUC scores indicate that our method effectively models *order sensitivity*.

**Robustness:** We randomly selected 1,000 image pairs (Appendix C). Our method outperformed across all findings in terms of the F1 score, demonstrating the robustness of our approach.

**Performance-Label Quality Alignment:** Enhanced performance after label curation demonstrates that our model's improvements mirror label quality. While consolidation shows less alignment, the marginal effect likely stems from the small number of samples impacted by curation (Appendix D).

## 4. Conclusion

We propose a straightforward bidirectional training framework for temporal CXR image classification, explicitly enforcing order sensitivity by inverting both image sequences and their labels. Evaluations on both internal and external datasets demonstrate consistent improvements—particularly in F1 score—underscoring the efficacy of our approach for real-world clinical scenarios.

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

## Appendix A.  LLM Label Extraction Detail

Figure 1: Label Space for Temporal Change Classification in Paired CXRs

In prior study, temporal image classification was typically framed as a 3-class problem—*worsened*, *stable*, or *improved*. However, we found this coarse-grained categorization insufficient to fully capture the spectrum of temporal changes in serial images. To address this, we designed a more fine-grained 5-class labeling scheme. As illustrated in Figure 1, the label space is partitioned based on both the progression and the existence of each finding.

For each radiological finding, Gemini-2.0 Flash assigned the label *New* when the finding was absent in the previous report but present in the current one, and *Resolved* when it was present in the previous report but no longer visible in the current report. The label *Worsened* was applied when the finding appeared in both reports with clear evidence of progression (e.g., increased size or severity). *Stable* was assigned when there was no appreciable change between the two reports. *Improved* was assigned when clear evidence of regression or clinical improvement was given. All analyses involving the datasets using the LLM were conducted in strict compliance with PhysioNet and relevant licensing guidelines.

This design ensures that each category occupies a comparably sized logical region, promoting a more balanced label distribution. The clear semantic boundaries between labels make label inversion not only feasible but also logically consistent. Furthermore, objectivity of the labeling criteria enables subsequent validation by human experts, thereby reinforcing the reliability of the labels generated by LLMs.

## Appendix B.  Order Sensitivity Validation Analysis

To quantify the model's awareness of input order, we conducted a order sensitivity validation using MS-CXR-T dataset. As shown in Table 3, the baseline model often predicts the same label (e.g., Worsen remains Worsen, Improved remains Improved) even when the input order is reversed—indicating a lack of sensitivity to temporal direction. In contrast, the bidirectional model shows a noticeable reduction in such cases, with predictions more

Table 3: Frequency table for original vs. reversed input sequences. Counter-diagonal entries indicate successfully inverted prediction pairs (e.g., *Worsen → Improved* when input order is reversed). O.P indicates prediction with original input sequence, R.P indicates prediction with reversed input sequence.

| Method | R.P / O.P | P. Effusion | | | Edema | | | Pneumonia | | | Pneumothorax | | | Consolidation | | |
|---|---|---|---|---|---|---|---|---|---|---|---|---|---|---|---|---|
| | | Worsen | Stable | Improved | Worsen | Stable | Improved | Worsen | Stable | Improved | Worsen | Stable | Improved | Worsen | Stable | Improved |
| **Baseline** | Worsen | 103 | 50 | **65** | 51 | 10 | **51** | 198 | 0 | **5** | 13 | 1 | **6** | 72 | 15 | **15** |
| | Stable | 50 | **53** | 0 | 9 | **8** | 6 | 0 | **0** | 0 | 1 | **16** | 5 | 20 | **30** | 1 |
| | Improved | **72** | 2 | 16 | **83** | 7 | 41 | **28** | 0 | 6 | **10** | 4 | 155 | **33** | 1 | 14 |
| **Unidirectional** | Worsen | 68 | 50 | **21** | 21 | 8 | **30** | 221 | 0 | **2** | 0 | 0 | **0** | 16 | 5 | **6** |
| | Stable | 29 | **187** | 8 | 9 | **56** | 24 | 0 | **0** | 0 | 0 | **104** | 27 | 11 | **142** | 2 |
| | Improved | **40** | 2 | 6 | **63** | 26 | 29 | **13** | 0 | 1 | **0** | 18 | 62 | **11** | 3 | 5 |
| **Bidirectional** | Worsen | 15 | 7 | **42** | 25 | 15 | **34** | 76 | 4 | **26** | 7 | 4 | **6** | 5 | 3 | **10** |
| | Stable | 6 | **227** | 15 | 20 | **74** | 13 | 2 | **21** | 0 | 1 | **139** | 7 | 2 | **139** | 5 |
| | Improved | **67** | 20 | 12 | **70** | 8 | 7 | **89** | 1 | 18 | **9** | 6 | 32 | **18** | 15 | 4 |

consistently flipping in accordance with the reversed input. Additionally, the proposed ICR metric can be computed as the sum of anti-diagonal elements divided by sum of total elements in the table.

## Appendix C. Dataset Label Distribution

| Findings | # of Labeled Pairs | New(1) | Worsen(2) | Stable(3) | Improved(4) | Resolved(5) |
|---|---|---|---|---|---|---|
| P. Effusion | 57127 | 27% | 15% | 27% | 16% | 15% |
| Edema | 37485 | 20% | 15% | 24% | 24% | 17% |
| Pneumonia | 8522 | 44% | 10% | 12% | 10% | 23% |
| Pneumothorax | 11165 | 20% | 11% | 29% | 14% | 26% |
| Consolidation | 29136 | 21% | 18% | 26% | 16% | 19% |

Table 4: MIMIC(5-class) Label Distribution

As shown in Table 4, most findings demonstrate a relatively balanced label distribution. However, pneumonia exhibits a notable class imbalance, with the *New* label comprising over 40% of the samples. This observation suggests that, for pneumonia in particular, evaluation metrics that are more robust to class imbalance—such as the F1-score—may be more appropriate than accuracy or other standard metric.

| Findings | # of Labeled Pairs | New(1) | Worsen(2) | Stable(3) | Improved(4) | Resolved(5) |
|---|---|---|---|---|---|---|
| P. Effusion | 59800 | 24% | 13% | 45% | 11% | 6% |
| Edema | 44746 | 17% | 13% | 44% | 19% | 7% |
| Pneumonia | 2011 | 58% | 11% | 17% | 7% | 7% |
| Pneumothorax | 17506 | 18% | 10% | 37% | 11% | 24% |
| Consolidation | 24342 | 19% | 15% | 48% | 10% | 8% |

Table 5: CheXpert(5-class) Label Distribution

In Table 5, for findings other than pneumonia, approximately 40% of samples were biased toward the *Stable*(3) label. Similar to the MIMIC dataset, pneumonia exhibited a smaller sample size and a skewed distribution toward the *New* (1) label.

## Appendix D. Error Analysis for Consolidation

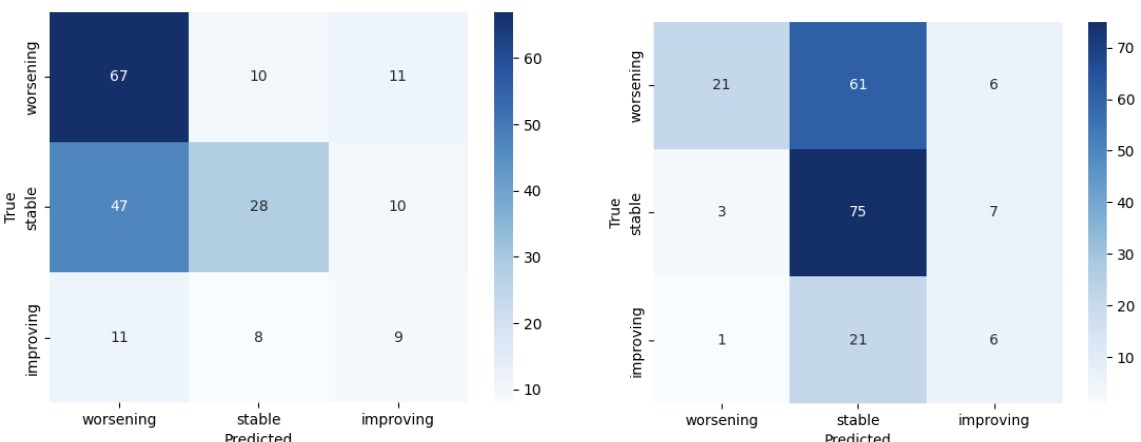

Figure 2: Confusion Matrix of **Baseline** method with Consolidation finding, MS-CXR-T

Figure 3: Confusion Matrix of **Bidirectional** method with Consolidation finding, MS-CXR-T

| Findings | # of Labeled Pairs | Worsening | Stable | Improving |
|---|---|---|---|---|
| P. Effusion | 1045 | 31.8% | 48.6% | 19.4% |
| P. Effusion + curation | 913 | 30.4% | 48.3% | 21.1% |
| Consolidation | 1045 | 43.7% | 42.2% | 13.9% |
| Consolidation + curation | 1013 | 44.3% | 42.0% | 13.6% |

Table 6: Label Distribution of Curated Sample

Figure 2, 3 present the confusion matrices for consolidation, comparing the baseline model (left) and our proposed method (right) on the MS-CXR-T dataset. While our method improves AUC, a slight drop in F1-score is observed. A closer examination of the confusion matrices provides insight into this trade-off.

Whereas the baseline model tended to over-predict *worsening*, the proposed method shifts this bias toward *stable*, leading to under-detection of actual *worsening* cases. This change reflects a more conservative prediction pattern, which improves overall calibration (as seen in higher AUC) but harms recall for critical progression classes. The ambiguous and overlapping radiographic features of consolidation may have contributed to increased confusion with the *stable* class.

Regarding the effect of curation for consolidation, the impact of curation appears minimal, given that the curation does not significantly change the label distribution as shown in Table 6.

