# OpenReview forum: "A Bidirectional Loss Approach to Imparting Order Sensitivity to Multi-Image Chest X-ray Encoders"
_MIDL.io/2025/Short_Papers — MIDL 2025 - Short Papers_

### Official Review · Reviewer_UKcT · 2025-04-28

**Rating:** 1
**Confidence:** 3

**Summary:**

The paper proposes a method to generate labels and augment a dataset of longitudinal changes in chest X-rays. It uses a large language model to extract labels representing the type of change between two reports of the same patient (resolved, improved, stable, worsened, and new). The authors train a classifier that receives two images in a certain order and outputs the class of longitudinal evolution between the two images. To ensure that the classifier considers the order that the two images are provided, for each pair of images, there is a loss term for their original order and an additional loss term for their inverted order and label (resolved inverts to new, improved inverts to worsened, and stable remains the same). The paper shows that a model trained of such a loss improves the overall classification scores on real cases in most scenarios.

**Strengths:**

-	The proposed label inversion method is simple yet likely leads to a significant improvement in final classification results and could become a standard technique for this type of task following the publication of this paper.

**Weaknesses:**

- The paper does not specify how authors process reports from the MIMIC-CXR and MS-CXR-T datasets without breaching their data user agreement that does not allow for sharing the data with a third party. Given that I only know of ways of using the Gemini-2.0 Flash that share data with Google, I believe that the paper is not respecting the data user agreement of those public datasets and should be rejected. An exchange of messages with the authors could potentially clear up the issue, but the review process of short papers from MIDL does not allow that. If it wasn’t for this fact, I believe the paper should be accepted (“weak accept”)

Minor
- The first sentence in the “Consistent F1 Gains:” paragraph in page 3 is confusing. I believe an “and” is missing.
- There is no discussion of the impact of the inverted labels into the causality of the cases. For example, is it okay for a case where first image is a lung with pneumothorax, second image is a lung with no pneumothorax but a chest tube, with a label “resolved” to be inverted?

---

### Decision · Program_Chairs · 2025-05-01

**Decision:**

Accept

**Comment:**

The PC discussed the paper during the panel meeting and decided to accept. However, please address the concerns of the reviewer regarding responsible use of the datasets in the camera-ready version. For example, did the authors follow the guidelines here: https://physionet.org/news/post/gpt-responsible-use